# Higher Sedentary Behaviors and Lower Levels of Specific Knowledge Are Risk Factors for Physical Activity-Related Injuries in Saudi Adolescents

**DOI:** 10.3390/ijerph20054610

**Published:** 2023-03-05

**Authors:** Mohamed Ahmed Said, Amnah Ahmed Almatar, Mohammed Shaab Alibrahim

**Affiliations:** 1Department of Physical Education, College of Education, King Faisal University, Al-Ahsa 31982, Saudi Arabia; 2Higher Institute of Sport and Physical Education of Kef, Jendouba 7100, Tunisia

**Keywords:** adolescents, bruises, strains, fractures, fat-free mass, limbs, medical attention

## Abstract

Encouraging physical activity (PA) for adolescents is necessary to achieve and maintain optimal health, but it may increase the risk of PA-related injuries. This study sought to assess the frequency, location, type, and severity of PA-related injuries in Saudi students aged 13 to 18 years and to identify associated risk factors. A total of 402 students, including 206 boys aged 15.87 ± 1.69 years and 196 girls aged 15.83 ± 1.70 years, were randomly assigned to participate in this study. For each participant, height, weight, body mass index, and fat percentage were measured. Responses to a specially designed four-part self-administered questionnaire were also collected. Results revealed that better specific knowledge was associated with a lower likelihood of sustaining injuries (β = −0.136; *p* = 0.001) but increased sedentary behaviors were associated with a greater likelihood of sustaining a PA-related injury (β = 0.358; *p* = 0.023). Gender, knowledge, and sedentary behaviors were the factors overall associated with a greater likelihood of suffering 1, 2, and 3 or more PA-related injuries. However, gender, fat-free mass, knowledge, and sedentary behaviors were associated overall with a greater likelihood of bruises, strains, fractures, sprains, concussions, and at least two types of PA-related injuries. Collectively, we should pay attention to the PA-related injury problem among middle and high school students, particularly when promoting a physically active lifestyle.

## 1. Introduction

Physical activity (PA) participation helps maintain and improve a person’s physical, social, and mental health at different stages of life, including increasing cardiorespiratory and muscular fitness, bone density, and reducing the risk of disabilities and communicable diseases [1]. The American College of Sports Medicine and the American Heart Association Physical Activity and Public Health Guidelines emphasize the importance of healthy adults getting at least thirty minutes of moderate-intensity physical activity daily, five days a week, or vigorous aerobic exercise for at least twenty minutes a day at least three times a week to prevent sudden cardiac death, cardiovascular disease, high blood pressure, stroke, osteoporosis, diabetes mellitus, metabolic syndrome, obesity, thirteen types of cancer, depression, functional health, and falls in the elderly [2].

Studies have also indicated that children and adolescents need at least an hour of physical activity of varying intensity most days of the week to promote health and well-being [2]. With the contemporary focus on a physically active lifestyle, an increased number of PA-related injuries can be expected [3]. Previous studies have found that PA-related injuries can be a heavy burden for children and adolescents, resulting in the absence of sports and normal PA engagement, physical discomfort, medical consultation, and hospitalization [4,5]. PA practice leads to the appearance of some external, internal, and endogenous factors of a biomechanical or physiological–biochemical nature affecting various systems of the body and leading to damage to the tissues of the trauma site and defect homeostasis [6]. Cai et al. [7] reported that although PA-related injuries are not seriously life-threatening, they have been ranked in the top half of treated injuries in school-aged adolescents in many countries, with nearly 39% of fracture cases resulting from sports and recreational activities. Therefore, effective injury prevention in school-aged children has great potential for public health gains. Two broad categories of PA-related injuries should be distinguished: acute and chronic. Acute injuries occur suddenly as a result of a fall, attack on an opponent, or joint strain, while chronic injuries develop gradually over time due to overuse or misuse of a muscle or group of muscles. Sprains and dislocations are examples of acute injuries, while tennis elbow and stress fractures are common chronic injuries [8].

Estimating the rate of PA-related injuries is challenging [1], based on risk factors and consequences. Many contributing factors can influence the frequency, extent, and risks of injuries. These factors are related to the participants’ characteristics, the type of activity performed, and other factors related to environmental conditions such as inadequate sports equipment, soil, and climate [9]. Lifestyle behaviors also affect the level of risk, such as physical activity and sedentary behaviors (SB) [10]. Relevant epidemiological studies also indicated that children and adolescents are more susceptible to PA-related injuries, with an estimated two-thirds of injuries occurring in participants aged 5 to 24 years being treated under medical supervision [11]. Sprains, fractures, superficial injuries and contusions, and traumatic brain injuries, including concussions, are the most common PA-related injuries in children and adolescents [12].

Merkel [13] estimated the disinclination to exercise and sport among 15-year-olds to be between 70% and 80%, injuries were one of the main reasons for dropping out of the sport. Sallis et al. [14] reported an incidence of 1874 PA-related injuries in a retrospective cohort study of 3767 adolescents participating in college-level competitions, split between female students, 856 injuries (45.7%), and male students, with 1018 injuries (54.3%). The authors noted no significant differences in injuries per 100 participants/year between genders (52.5 for girls versus 47.7 for boys), except for swimming and water polo, where swimmers reported more back, neck, shoulder, thigh, knee, and foot injuries, while water polo players reported more shoulder injuries. When all activities were assessed at the same time, this study showed higher rates of hip, leg, and shoulder injuries in girls, while male athletes reported higher injury rates at the thigh.

In light of these observations, the development of a strategy aimed at reducing the risk of injuries, protecting individual health, and increasing physical and sports safety is a crucial element in reducing the reluctance of children and adolescents to engage in PA. Ensuring participation in PA contributes to the development of public health [15]. This strategy requires continuous monitoring of the spread of PA-related injuries, strengthening protective measures, and limiting the factors that cause them.

The assessment of PA characteristics is one of the methods used to identify possible modifiable and intrinsic risk factors for PA-related injuries. Several studies have shown a close relationship between the various components of physical fitness and risk factors associated with PA-related injuries [16], such as high body weight and its association with increased force absorption by soft tissues and joints [17]. Additionally, high body weight corresponds to decreased hip adductor muscle strength, which has been associated with increased hip injuries [18,19].

According to Truong et al. [19], education about PA-related injuries is one of the most significant factors in promoting a positive attitude toward treatment and prevention. The exchange of resources, such as informational (e.g., educational) or emotional support, positively influenced recovery expectations and risk assessment during rehabilitation or a return to sports. In addition, an injury role model, a forum for discussion and sharing of experiences, and the creation of new roles and social networks have been essential in improving prevention, staying motivated, and effectively managing injury recovery [20].

Epidemiological studies have shown that injured athletes value healthcare professionals who engage and involve them in their care through strategies such as goal-setting. A strong therapeutic alliance in which the goals and values of each athlete are respected results in positive rehabilitation experiences and increased trust in providers. Open discussions with providers and coaches allowed athletes to make decisions about surgery, rehabilitation, or a return to sports, helping prevent injuries. Beidler et al. [21] showed that participants’ lack of knowledge about PA-related injuries is likely to be a reason for the increased rate of PA-related injuries. Understanding the relationship between these two factors in the rehabilitation process could speed recovery and injury management [22].

Therefore, the present study aimed to assess the frequency of PA-related injuries among Saudi adolescents aged 13–18 to determine the location of the injury in the body area, the types of injuries, and whether the injury required urgent medical attention, and subsequently to investigate the association of these factors with gender, age, body mass index (BMI), knowledge of PA-related injuries, SB, and level of weekly physical activity. We hypothesized that boys reported more PA-related injuries than girls. We also hypothesized that participants with a good knowledge of PA-related injuries would be less likely to sustain PA-related injuries than participants with low knowledge. Additionally, it was hypothesized that participants with higher SB and reduced levels of PA were more likely to sustain PA-related injuries than those with lower SB and a higher level of PA. Finally, we hypothesized that lower limb bruises and strains were the most common PA-related injuries among Saudi boys, while upper limb strains were the most common PA-related injuries among girls.

## 2. Methods

### 2.1. Participants

According to the Saudi Ministry of Education—General Education Statistics at https://www.moe.gov.sa/en/knowledgecenter/dataandstats/pages/infoandstats.aspx# (accessed on 24 February 2023), in the 2022–2023 school year, the total number of students enrolled in public middle and high schools in Al-Ahsa governorate was 94,384 students, including 46,453 males and 47,931 females. Referring to Krejcie and Morgan’s table [23], to determine the necessary size of a randomly selected sample from a given finite population, the minimum sample size should be approximately 384 students. Considering a dropout rate of 20% or more [24] and the proximity of the number of boys and girls, a minimum of 500 students was required to participate in our study (boys: *n* = 250; girls: *n* = 250). To ensure this number, eight boys’ schools and eight girls’ schools were randomly selected, with two schools in each geographical area (east, west, north and south). In each school, one class at each grade level was randomly selected, with 10 to 20 students per class, and excluding members who met one or more exclusion criteria, all remaining students were invited to participate in this study. Exclusion criteria were (1) any amputation of any part of the body; (2) an acute or chronic medical condition; and (3) non-completion of one or more phases of the study. Lack of written parental consent for children’s participation in our study was also considered an exclusion criterion.

The positive responses received concerned 410 students (210 boys and 200 girls), while the others refused to participate. The disapproval was mainly attributed to cultural reasons that discourage participation in such studies. Before the measurements, students who met one or more criteria from the following list were excluded: age under 13 or over 18, disability, or any contraindication to PA. Failure of one or more of the study trials and lack of signed informed consent from participants aged 16 or older or parents younger than 16 were also considered exclusion criteria.

After measurements, the results were checked for outliers with gender-specific Z-scores. As a general rule, values with a Z-score greater than 3 or less than −3 are often considered outliers and were excluded. A total of 8 outliers were determined and the final sample retained for analysis was 402 students (206 boys and 196 girls) aged 13 to 18. Participants were categorized into four BMI groups according to age- and sex-specific cutoffs suggested by the 2007 WHO Growth Reference (5–19 years): (1) underweight (UW; BMI-for-age < 5th percentile), normal weight (NW; 5th percentile ≤ BMI-for-age < 85th percentile), overweight (OW; 85th percentile ≤ BMI-for-age < 95th percentile), and obese (OB; BMI-for-age ≥ 95th percentile) (from de Onis et al. [25]).

### 2.2. Procedures and Ethics

This study was conducted between September 2022 and November 2022 in accordance with the Declaration of Helsinki (rev. 2013) relating to the ethical principles for medical research involving human subjects [26]. All procedures conducted were approved by the Ethics Committee of the Deanship of Scientific Research, King Faisal University (Ref. No. KFU-REC-2021-OCT-EA00019).

Prior to the measurements, participants and their parents were invited to a meeting with the research team to be informed about the purpose of the study and to understand the study protocol and possible effects that might result. An explanatory letter was also sent to all parents who were unable to attend the meeting and who were asked to complete an informed consent form for their children’s participation in the study. Once the ethical consents were signed, the height, weight, and body composition were measured for each participant, and at the same time, responses to a specially designed self-administered four-part questionnaire were collected (demographics, knowledge of PA-related injuries, PA levels, and SB). All measurements were performed in air-conditioned classrooms and all participants were instructed not to eat or drink two hours before the anthropometric measurements and to dry their feet thoroughly before walking on the measuring device.

### 2.3. Measurements

#### 2.3.1. Anthropometric Parameters

For each participant, height was measured using a stadiometer (Holtain, Crymych, Wales, UK), and weight, BMI, and fat percentage were determined using bioelectrical impedance with a body composition monitor (Omron, Body Composition Monitor BF 508, Kyoto, Japan).

#### 2.3.2. Physical Activity-Related Injuries Parameters

Based on a questionnaire from the work of Wang and Huang [27] and Pi and Yeh [28], the physical activity-related injury parameters were translated into Arabic by Aldaferi et al. [29]. The objective of the questionnaire was to determine the prevalence of injuries related to PA and the level of knowledge related to this type of injury among the participants. The total number of questions was 30, of which the first 5 questions were designed to assess PA-related injuries that participants remembered and reported from the previous year, while the remaining 25 questions assessed the knowledge of PA-related injuries. All questions were multiple choice and the definition of PA-related injury as described by Bloemers et al. [30] was applicable to this study.

An acceptable PA-related injury must have met at least one of the following criteria: the student (1) must stop the current PA; (2) cannot participate in the next scheduled PA; (3) cannot go to class the next day; (4) should seek medical attention (including first aid, medical consultation, or physical therapy, but excluding those using only bandages) [3]. Questionnaire items regarding PA-related injuries occurrences and reports were worded as follows: (1) “During the past year, have you suffered a physical activity-related injury?” Two options were provided: “yes” or “no”; (2) “How many physical activity-related injuries do you think you have experienced in the last year?” Four options were provided: “no”, “once”, “twice”, and “three or more times”; (3) “What type(s) of physical activity-related injury(ies) did you have in the last year?” Five options were provided: “bruises”, “strains”, “fractures”, “sprains”, and “two or more types”; (4) “In which part(s) of your body did physical activity-related injury(ies) occur?” Four options were provided: “head”, “trunk”, “upper limbs”, and “lower limbs”; and (5) “Have you had any physical activity-related injuries in the past year that required urgent medical attention?” Two options were provided: “yes” or “no”.

Participant knowledge was assessed with 25 five-point Likert-scale questions concerning symptom recognition, complications, and general knowledge about PA-related injuries and their first aid. The correct answer for each question was identified and coded 1, while the wrong answer was coded 0. The total score for knowledge was determined by adding the scores of the 25 questions. Scores ranged from 1 to 25, meaning that the higher the score, the higher the knowledge. Using three percentiles (33) to determine knowledge of PA-related injuries, a score of 1 to 8 points indicates poor proficiency, a score of 9 to 16 points indicates intermediate proficiency, and a score of 17 or more points indicates good proficiency.

#### 2.3.3. Parameters Related to Physical Activity

The assessment was conducted using the Arabic version of the Physical Activity Questionnaire for adolescents (PAQ-A) [31]. PAQ-A is a seven-day self-administered questionnaire designed to assess participation in various types of PA, including activities during physical education sessions, lunch breaks, after school, evenings, and weekends. This questionnaire was originally designed for high school students in grades 9 to 12 and approximately 14 to 19 years of age. It consists of nine (9) items where items 1 through 8 were scored from 1 to 5 on a 5-point Likert scale. A mean composite score for the PAQ-A was then calculated for items 1–8, excluding item 9, where a score of 1 indicates a low level of PA and a score of 5 indicates a high level of PA. The average score made it possible to classify the participants into five categories: highly inactive (score < 1.8), inactive (1.8 to <2.6), moderately active (2.6 to <3.4), active (3.4 to <4.2), and very active (4.2 or over).

To be easily understood by all participants, the English version of the PAQ-A has been translated and adapted to the Saudi context following the recommendations of Beaton et al. [32]. The original version of the PAQ-A was translated into Arabic by two freelance translators who are fluent in English but whose mother tongue is Arabic. The initial translation was then back-translated by two other freelance translators to ensure accuracy. The two versions of the translations were compared and a final compatible version was agreed upon. It was evaluated by specialists in physical education and health sciences, in terms of unequivocal understanding of the content of the questions, transparency of all the points, and their usefulness. The accepted version of the questionnaire was then back-translated into English by two independent translators who were unfamiliar with the original version of the PAQ-A and whose mother tongue was English (native speaker). Finally, the necessary adjustments requested by the experts were made and the validity and reliability of the Arabic version were tested in a pilot sample of 47 participants.

#### 2.3.4. Parameters Related to Sedentary Behaviors

Evaluated using the NSW Schools Physical Activity and Nutrition Survey [33], the survey was translated and adapted to the Saudi context by Said et al. [34] and contained four entries on sedentary activities involving the use of technological devices and the Internet (iPad, tablet, computer, or smartphone), watching television or films or programs on the Internet, and playing games (including on computers, game consoles, smartphones, or iPads). Possible answers were: less than 30 min per day, scored 5; 30 min^−1^ h daily, rated 4; about 1–2 h per day, rated 3; about 2 to 4 h per day, rated 2; and more than 4 h per day, rated 1. An average score between 1 and 5 was calculated for each participant, with a higher score indicating more SB and a lower score indicating less SB [35].

#### 2.3.5. Reliability and Validity of PAQ-A

The exploratory factor and principal component analysis results indicate that the Kaiser–Meyer–Olkin measure of sampling adequacy was 0.697, which is above the acceptable 0.6 threshold. The Bartlett’s test sphericity score was statistically significant (*p* < 0.001), indicating that all commonalities were greater than 0.30. The item scores were then subjected to an exploratory factor analysis, which identified three components relating to the Arabic version of the PAQ-A equal to 55.762% of the total variance, confirming that the eight elements represented a three-dimensional construct. The load factor for all items indicates that all items in the domain made a significant contribution (>0.50: 0.591–0.811). Next, the Arabic version of the PAQ-A was tested for reliability using Cronbach’s alpha; the results indicate acceptable internal consistency (0.69). In addition, all items achieved an acceptable corrected item-total correlation of >0.30. Thus, the Arabic version of the PAQ-A instrument was valid and reliable for measuring the PA levels of Saudi students in grades 9 to 12.

### 2.4. Statistical Analysis

The descriptive analysis of the sample and variables was conducted with the calculation of means, standard deviations, and frequencies. The normality of the distributions was tested using histograms and absolute values of skewness and kurtosis. An absolute skew value greater than 2 or an absolute kurtosis greater than 7 were used as reference values to determine substantial non-normality [36]. Student’s t-test was used to identify statistical differences between independent groups stratified by dichotomized variables (gender and response to questions 1 and 5). The magnitude of differences between boys and girls in terms of height, weight, and body composition was measured using Cohen’s d, with 0.2, 0.5, and 0.8 representing, respectively, low, medium, and high effect sizes. One-way ANOVA was used for subsequent comparisons of k-independent samples stratified by responses to questions 2, 3, and 4. For significant results and due to unequal sample sizes, the Tukey–Kramer method was used for doing post hoc comparisons. A binary logistic regression (logit) model was applied to analyze the independent variables with the dichotomous variables (questions 1 and 5 (1 = yes; 0 = no)), with age and gender as categorical variables. Multinomial logistic regression was used to test the effects of independent variables on dependent variables with more than two categories (question 2 (0 = Not injured; 1 = Once; 2 = Twice; 3 = Three or more times), question 3 (0 = Not injured; 1 = Bruise; 2 = Strains; 3 = Fractures; 4 = Sprains; 5 = Concussions; 6 = Two or more types), and question 4 (0 = Not injured; 1 = Head; 2 = Torso; 3 = Upper limbs; 4 = Lower limbs; 5 = Two or more parts)), the first category (Not injured) served as a reference-category in all testing and the variance inflation factor (VIF) was used to test for multicollinearity. Statistical analysis was performed using SPSS V.26 (IBM, Armonk, NY, USA) and *p*-values were set to 0.05.

## 3. Results

### 3.1. Sample Characteristics

A total of 402 students completed all the parts of the study and were analyzed. Of these, 196 were girls (48.76%) aged 15.83 ± 1.7 years, and 206 were boys (51.24%) aged 15.87 ± 1.69 years. Referring to the sex-specific BMI-for-age percentile charts, the overall prevalence of overweight and obesity in boys was 11.7% and 29.1% and in girls 8.7% and 15.3%, respectively. Significant differences were noted in weight, height, BMI, PBF, and fat-free mass (*p* = 0.004 for BMI; *p* = 0.001 for the rest) between groups of boys and girls. However, Cohen’s relative d-values for BMI, PBF, and FM were less than 0.5, demonstrating that differences between boys and girls in these variables were so limited. No significant sex differences were noted in age and fat mass (Table 1).

Of the 402 individuals surveyed, 184 participants (45.77%; 118 males and 66 females) reported having suffered PA-related injuries in the past year, of which 93 subjects (55 males and 38 females) suffered one injury, 54 subjects (41 males and 13 females) suffered two injuries, and 37 subjects (22 males and 15 females) suffered three or more PA-related injuries. Bruises (29 cases) and strains (27 cases) were the most common types of PA-related injuries among male students; conversely, bruises (16 cases) and fractures (10 cases) were the most prevalent PA-related injuries among female students. Fifty-nine participants (36 males and 23 females) reported experiencing two or more types of PA-related injuries. The limbs were the most injured parts of the body in boys (18 cases for upper limbs and 79 cases for lower limbs), while the head and lower limbs were the most vulnerable parts in girls (respectively, 12 and 34 cases). Of all the injuries that occurred, only 43 cases (33 boys and 10 girls) required urgent medical intervention.

### 3.2. Knowledge of PA-Related Injuries

Participants’ average correct answer score was 10.31 ± 3.68, with 10.57 ± 3.97 for boys and 10.03 ± 3.34 for girls. The analysis of the individual values shows that 33.3% of respondents had a low competence level (score of 1 to 8 points), 61.7% had an intermediate competence level (score of 9 to 16 points), and only 5% of respondents had a good level of competence (at least 17 points).

The one-way ANOVA test showed that levels of knowledge about PA-related injuries were almost similar across all age groups. However, significant differences were noted among participants stratified by BMI categories (*f* = 4.302; *df* = 398; *p* = 0.005), demonstrating higher competence among overweight participants compared to their normal-weight (*p* = 0.002) and obese (*p* = 0.027) peers. Higher competence was also noted in uninjured students than in injured peers (*p* = 0.001; Table 2) and in injured twice than those injured three or more times (*p* = 0.047; Figure 1A), and in students with lower limb injuries compared to those with upper limb injuries (*p* = 0.039; Figure 1C). No significant differences were noted between subgroups stratified by gender, type(s) of injury (Figure 1B), and need for urgent medical intervention (Table 2 and Table 3).

### 3.3. Sedentary Behaviors

The mean SB score for participants was 2.716 ± 0.61, with 2.79 ± 0.596 for boys and 2.64 ± 0.67 for girls (*p* = 0.011). The *t*-test revealed that injured students were significantly more sedentary than their uninjured peers (*p* = 0.043), and those who needed medical intervention than those who did not need it (*p* = 0.026). Among injured students, the one-way ANOVA revealed significant differences between groups stratified by the number of injuries (*f* = 5.032; *df* = 181; *p* = 0.007; Table 2), with significantly higher SB among participants injured three or more times than among those injured only once in the past year (*p* = 0.012; Figure 2A). No significant differences were noted between subgroups stratified by injury type or injured body part (Figure 2B,C).

### 3.4. Weekly Physical Activity

The average PA score of the participants was 2.45 ± 0.86 (boys = 2.32 ± 0.9; girls = 2.58 ± 0.8) indicating that our participants ranged from inactive to moderately active. The *t*-test revealed that injured students were as active as their uninjured peers, but those with injuries that required medical attention were less active than those with minor injuries (*p* = 0.019). Among injured students, significant differences were also noted between groups stratified by number of injuries (*f* = 3.118; *df* = 181; *p* = 0.047) and injured body parts (*f* = 3.032; *df* = 180; *p* = 0.019; Table 2). The post hoc test revealed that those injured three or more times were significantly less active than those injured only once in the past year (*p* = 0.034; Figure 3A). A significant difference was also found in favor of participants injured in the lower limbs versus those injured in two or more parts of the body (*p* = 0.004; Figure 3C). No significant differences were noted between subgroups stratified by lesion types (Figure 3B).

### 3.5. Regressions

Logistic regression was performed to determine the effects of gender, age, BMI, fat mass, lean mass, knowledge of PA-related injuries, SB, and level of PA on the likelihood of having PA-related injuries in general and the likelihood that the injury will require emergency medical attention. The logistic regression model was statistically significant, χ^2^(12) = 55.416, *p* = 0.001. The model explained 17.2% (Nagelkerke R^2^) of PA-related injuries variance and correctly classified 64.7% of cases. Boys were 3.52 times more likely to have PA-related injuries than girls. Increased knowledge was associated with lower odds of experiencing PA-related injuries, but increased SB was associated with increased odds of experiencing PA-related injuries.

The logistic regression model was also statistically significant for the need for medical care after PA-related injury, χ^2^(12) = 26.015, *p* = 0.011. The model explained 12.7% (NagelkerkeR^2^) of the variance in the degree of PA-related injuries needing medical care and correctly classified 89.3% of cases. Boys were 2929 times more likely to be severely injured, requiring emergency medical intervention than girls. Increased SB increased the likelihood of needing medical attention for PA-related injuries (Table 4).

Analysis of dependent variables with more than two categories (questions 2, 3, and 4) was performed using multinomial logistic regression with gender, age, BMI, fat mass, lean mass, knowledge of PA-related injuries, SB, and level of PA as independent variables and “Not injured” as the reference category. Pearson’s chi-square statistic indicated that the model fitted well with data on the number of PA-related injuries sustained, χ^2^(1167) = 1180.615, *p* = 0.384. Likelihood ratio tests also indicated that gender, knowledge of PA-related injuries, and SB were the independent variables overall associated with a greater likelihood of sustaining 1, 2, and 3 or more PA-related injuries. As the dependent variable has four categories and due to the second set of coefficients not being significant, only two logits were retained, indicating that boys with low knowledge of PA-related injuries were more likely to experience PA-related injuries than not; Exp(β) values were 4.207 and 0.876, respectively. Multinomial regression results also showed that 14- and 16-year-olds with low knowledge associated with high SB were more likely to have three or more PA-related injuries than none (Table 5).

Regarding types of PA-related injuries, Pearson’s chi-square statistic indicated that the model fits the data well, χ^2^(2334) = 2190.1, *p* = 0.984. Likelihood ratio tests also indicated that gender, fat mass, fat-free mass, knowledge of PA-related injuries, and SB were the independent variables overall associated with a greater likelihood of sustaining bruises, strains, fractures, sprains, concussions, and two or more types of PA-related injuries. The model coefficients indicated that a boy was more likely to have bruises, strains, and fractures 7024, 5447, and 9023 times, respectively, than to remain unharmed. Additionally, reduced lean body mass was more likely to increase the risk of bruising (Exp(B) = 0.933) and fractures (Exp(B) = 0.883); however, increased SB was more likely to increase the risk of having strains (Exp(B)= 2.001), concussions (Exp(B) = 3.936) and two or more injury types (Exp(B) = 1.666). Finally, reduced knowledge of PA-related injuries was likely to increase the likelihood of having bruises (Exp(B) = 0.837), sprains (Exp(B) = 0.854), concussions (Exp(B) = 0.793), and two or more injury types (Exp(B) = 0.869), instead of not being injured.

Regarding the injured body parts, fit testing indicated that the model fit the data well, χ^2^(1556) = 1571.687, *p* = 0.385. Likelihood ratio tests also indicated that gender, knowledge of PA-related injuries, and level of PA were the independent variables globally associated with a greater likelihood of developing PA-related injuries in the head or trunk, upper limbs, lower limbs, and two or more parts of the body. The model coefficients indicated that participants were more likely to sustain head or trunk injuries if they were 16 or 17 years old and had low knowledge of PA-related injuries. Additionally, rather than remaining unharmed, Saudi students were more likely to sustain upper extremity PA-related injuries (1) if they were males, had low knowledge, and had high SB; (2) lower limbs if they were male, 15 years old, and had low knowledge; and (3) two or more body parts if they were 16 years old and had low knowledge of PA-related injuries and weekly PA.

## 4. Discussion

The purpose of this study was two-fold: (1) to assess the frequency of PA-related injuries among Saudi students aged 13–18 to determine the location of the injury in the body area, the types of injuries, and whether or not the injury required urgent medical attention, and (2) to investigate the association of these factors with gender, age, BMI, the student’s extent of knowledge about PA-related injuries, level of SB, and their weekly PA.

Results revealed that boys were at higher risk for injury than girls, 57.28% of boys and 33.67% of girls reported having suffered PA-related injuries over the past year. Recorded values were greater than those noted by Cai et al. [7] in middle school students (25.1%); Räisänen et al. [37] in children and adolescent sport club activities, leisure time PA, and school-based PA (46%, 30%, and 18%, respectively); and Tang et al. [3] in middle-school students (33.6%, with 42.0% for boys and 25.0% for girls). However, findings were lower than those reported by Ristolainen et al. [38] in 15- to 35-year-old cross-country skiers, swimmers, long-distance runners, and soccer players (92% of males and 79% of females), and those reported by von Rosen et al. [39] in elite adolescent athletes (91.6%).

According to our findings, three factors might contribute to the elevated overall incidence and gender discrepancy: first, a low overall level of PA was reported among participants, more in boys than girls; second, a reduced knowledge of PA-related injuries was seen among participants; and third, a higher SB was reported in boys compared to girls. The difference in fat-free mass between boys and girls and the maturity stage might also contribute partly to the occurrence of some types of PA-related injuries.

Note that these results were expected and reasonable given the limited access of Saudi women to sports and PA. Indeed, various social and religious conditions have made women’s sports in Saudi Arabia a controversial topic for many years due to the suppression of women’s participation in sports by conservative authorities. Saudi women were not allowed to play sports in public places, go to school alone, or engage in recreational activities with or in front of men. The first sports center dedicated to girls in Saudi Arabia was established in 2013, offering training programs including fitness, karate, yoga, weight loss, and special activities for children.

Unlike boys, physical education has just been introduced into the education system for girls, and it is not yet implemented in all schools. The physical education program in girls’ schools was approved in 2017 with the aim of boosting sports activities among all members of Saudi society, one of the goals of the Kingdom’s Vision 2030. In a recent systematic review of the prevalence of inactivity and perceived barriers to active living, Al-Hazzaa [40] stated that the most important barriers to PA among adolescents are lack of time among both males and females, followed by lack of appropriate location, especially among females, and lack of facilities and resources. The dry, tropical climate, hot in the summer and windy in the winter, is also likely to limit the practice of sports, particularly outdoor sports. Haphazardly placed community facilities in neighborhoods also do not facilitate daily walking or biking due to the limited number of health trails and narrow, poorly designed sidewalks [41]. Collectively, we should focus adequate attention on the PA-related injury problem among middle and high school students, particularly when promoting a physically active lifestyle [3,7].

Results also demonstrated that bruises and strains were the most common types of sports injuries among boys, while bruises and fractures were the most common injuries among girls. Among subjects, 32.07% of the injured participants reported having suffered at least two types of injuries. The limbs were the most injured body parts in boys, while the head and lower limbs were the most vulnerable parts in girls. Regarding the injuries, 23.37% required urgent medical attention. Participants aged 15 to 17 years were more vulnerable to repeated injuries (three or more) in various parts of the body (two or more), mainly the head and trunk.

The growth and development of adolescents have implications for specific risks of sports-related injuries. Growth and development can also impact short- and long-term complications of sports-related injuries [42]. According to da Costa et al. [10], growth spurt, maturity-related variations, and lack of complex motor skills required for certain sports are some of the risk factors that can play an important role in the growth of the athlete. Pubertal changes occur in a predictable stepwise manner among which the asynchronous structural and tissue changes are responsible for the appearance of lesions [42,43]; as bones expand rapidly during adolescence, soft tissues slowly and passively stretch, becoming progressively tighter, and an imbalance between strength and flexibility can occur. At 15 and a half for girls and 17 for boys, the length of the trunk and legs is almost final while the muscles have not yet reached their maximum size—lack of strength can become a potential cause of injury. This can lead to abnormal movement mechanics and decreased motor performance during maximum speed at height [44]. Bone mineralization may also not be in line with its linear growth and therefore the bone is temporarily more porous, so there is an increased risk of fractures throughout the bone and growth plate [11,45]. All of these events, acting separately or together, render the immature musculoskeletal system less able to cope with traumatic situations and repetitive biomechanical stresses [10].

Knowledge of PA-related injuries is also a relevant factor in preventing the occurrence of injuries. The results found negative associations between knowledge of PA-related injuries and all variables related to injury frequency, location, and types, except bruises and strains compared to no injuries. However, knowledge of PA-related injuries cannot be studied independently of cognitive development in general. According to Brown et al. [42], teenagers can set more realistic goals for physical or athletic ability and participation. As decisions become more forward-thinking, conflicting priorities, including academics, appointments, and future professional or educational goals, can outweigh the importance of physical participation for an individual. At this stage in life, an adolescent’s personal values may be more clearly defined and his intellectual capacity, functional capacity, and abstract thought processes are well developed [46,47,48]. With respect to sports participation, adolescents now possess the cognitive ability to understand and memorize complex strategies and have fully developed perceptual motor skills. Skill enhancement is not limited to physical prowess, but includes mental capacity, situational awareness, awareness of hazards that may arise, and risk reduction. Adequate knowledge can help achieve good rehabilitation and become very important to improve recovery and prevent further injury [49]. In a study conducted on the prevention of elbow injuries in young baseball pitchers, Fleisig and Andrews [50] ultimately concluded that some elbow injuries in young baseball pitchers can be prevented through safety rules, recommendations, education, and common sense. The authors stated that young players need to be equipped with adequate knowledge about risk factors and safety guidelines so that they can fend for themselves. Knowledge in these areas would improve players’ ability to recognize early signs and symptoms of overuse injuries and confidently report them to coaches and parents. Additionally, this knowledge can enable players to actively participate in tracking the number of pitches and innings/games played, which in turn can promote better compliance by parents and coaches [51].

The data also revealed that SB were associated with a greater likelihood of developing a number and type of sports injuries: for every one-point increase in SB, the likelihood of being injured increased by 1.431, having an injury that requires medical intervention by 1.87, suffering three or more sports injuries by 2.274, having strains by 2.001, having concussions by 3.936, having two or more types of injuries by 1.666, and having an upper extremity injury by 2.302. Cairncross et al. [52], examining the association between initial screen time and post-concussion symptom severity in children and adolescents with a concussion, as compared to those with an orthopedic injury, demonstrated that screen time was a significant but nonlinear moderator of group differences in post-concussion symptom severity for parent-reported somatic symptoms and self-reported cognitive symptoms. Both low and high screen times were associated with relatively more severe symptoms in the concussion group than in the orthopedic injury group during the first 30 days after injury. The available data are not yet sufficient to determine a strong association between SB and health status in children and adolescents; however, the World Health Organization 2020 guidelines on physical activity and sedentary behavior affirmed that in children and adolescents, higher amounts of sedentary behavior are associated with detrimental effects on the following health outcomes: fitness and cardiometabolic health, adiposity, behavioral conduct/prosocial behavior, and sleep duration [53]. The association between sedentary behavior and adverse health outcomes is generally stronger for television or recreational screen exposure than for total sedentary time. The guidelines also noted that time spent in SB can include activities such as reading, studying, drawing, crafts, music, etc., and that these activities can have cognitive and other benefits [53]. Additionally, Panahi and Tremblay [54] argued that SB negatively affect appetite control, associated peripheral biomarkers, and neuro-messengers. According to these authors, there appears to be a modern version of sedentary behavior related to screening activities, which has a potential neurogenic component leading to overeating, stress, and adverse effects on metabolic health that indirectly contributed to the occurrence of wounds. Therefore, preserving movement for optimal health and implementing some of these strategies to increase participation in physical activity in schools and workplaces is extremely beneficial. Children and adolescents should limit the amount of time spent being sedentary, particularly the amount of recreational screen time [53].

Several limitations of this study should be considered. First, because our data are self-reported, recall and reporting biases are unavoidable. Students can clearly remember late and serious injuries but easily forget earlier and minor injuries [7]. Reference to the personal health passport provided by the Saudi Ministry of Health for injuries requiring a visit to the hospital or to the doctor could provide more reliable and diversified data. Second, direct assessment of PA and SB levels using pedometers or any other direct method will provide more reliable data. Finally, the existence of missing data for PA-related injury outcomes would, to some extent, negatively affect the severity analysis among injured students [3]. Integrating this study with data on sport participation and injury outcomes will provide valuable results for readers, professionals, and managers.

## 5. Conclusions

Findings revealed that boys were 3.52 times more likely to have PA-related injuries than girls. More specific knowledge was associated with a lower likelihood of sustaining a PA-related injury. In contrast, increased SB was associated with a greater likelihood of sustaining a PA-related injury. Bruises and strains were the most common types of sports injuries among boys, while bruises and fractures were the most common injuries among girls. The limbs were the most injured body parts in boys, whereas the head and lower limbs were the most vulnerable parts in girls. Of the injuries reported, 23.37% required urgent medical attention and participants aged 15 to 17 were the most vulnerable to repetitive injuries (three or more) in various body parts (two or more), primarily the head and the trunk.

## Figures and Tables

**Figure 1 ijerph-20-04610-f001:**
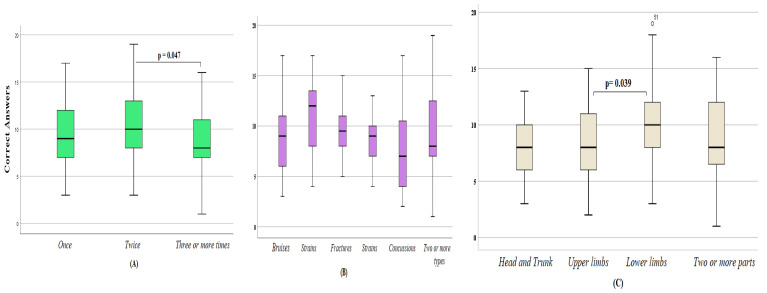
Correct answer score among Saudi adolescents aged 13–18, who suffered injuries in the last year, stratified by the number of injuries (**A**), type(s) of injury (**B**), and the injured part(s) of the body (**C**).

**Figure 2 ijerph-20-04610-f002:**
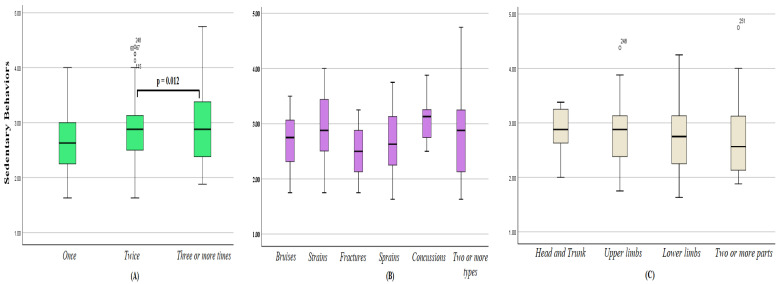
Sedentary behaviors score among Saudi adolescents aged 13–18, who suffered injuries in the last year, stratified by the number of injuries (**A**), type(s) of injury (**B**), and the injured part(s) of the body (**C**).

**Figure 3 ijerph-20-04610-f003:**
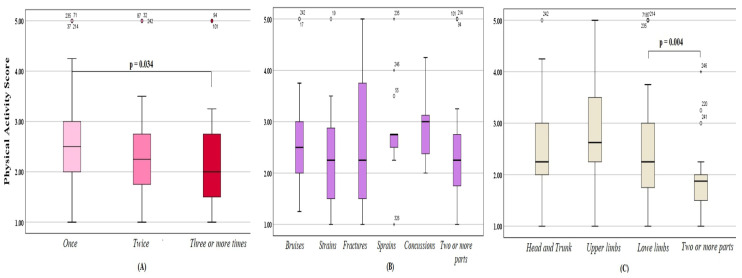
Physical activity score among Saudi adolescents aged 13–18, who suffered injuries in the last year, stratified by the number of injuries that occurred (**A**), type(s) of injury (**B**), and the injured part(s) of the body (**C**).

**Table 1 ijerph-20-04610-t001:** Anthropometric characteristics of Saudi students aged 13–18, stratified by gender.

		*N*	Mean	Std. Deviation	t	*p*	Cohen’s d
Age (years)	Boys	206	15.87	1.69	0.195	0.845	0.024
	Girls	196	15.83	1.70			
Weight (kg)	Boys	206	63.75	17.25	8.51	0.001	0.84
	Girls	196	50.81	13.18			
Height (cm)	Boys	206	169.32	8.04	18.65	0.001	1.86
	Girls	196	155.85	6.33			
BMI (kg·m^−2^)	Boys	206	22.32	5.47	2.92	0.004	0.29
	Girls	196	20.83	4.96			
PBF	Boys	206	23.80	10.35	−4.35	0.001	0.4
	Girls	196	28.50	10.95			
Fat mass (kg)	Boys	206	16.59	11.17	0.87	0.385	0.09
	Girls	196	15.75	10.16			
Fat-free mass (kg)	Boys	206	47.17	8.18	18.78	0.001	1.88
	Girls	196	35.06	3.95			

Data are means and standard deviation. BMI = body mass index; PBF = percentage body fat.

**Table 2 ijerph-20-04610-t002:** Knowledge of physical activity-related injuries, sedentary behaviors, and level of physical activity among Saudi students aged 13–18, stratified by sex, age, BMI categories, frequency, and types of injuries.

		*N*	Correct Answers	Sedentary Behaviors	Physical Activity Score
Gender	Total	402	10.31 ± 3.68	2.716 ± 0.61	2.45 ± 0.86
	Boys	206	10.57 ± 3.97	2.79 ± 0.596	2.32 ± 0.9
	Girls	196	10.03 ± 3.34	2.64 ± 0.67	2.58 ± 0.8
		t	1.492	2.545	−3.142
		*p*	0.137	0.011	0.002
Age (years)	13	52	10.81 ± 4.22	2.82 ± 0.56	2.38 ± 0.83
	14	51	10.61 ± 3.42	2.76 ± 0.70	2.51 ± 0.76
	15	56	10.84 ± 4.17	2.67 ± 0.58	2.48 ± 0.86
	16	77	10.04 ± 3.03	2.75 ± 0.65	2.34 ± 0.80
	17	79	9.81 ± 3.64	2.58 ± 0.57	2.57 ± 1.01
	18	87	10.17 ± 3.72	2.75 ± 0.57	2.41 ± 0.83
		F	0.88	1.333	0.738
		*p*	0.491	0.249	0.595
BMI Categories	Underweight	57	10.40 ± 3.310	2.695 ± 0.635	2.373 ± 0.916
	Normal weight	214	9.95 ± 3.808	2.659 ± 0.587	2.540 ± 0.878
	Overweight	41	12.17 ± 3.619	2.845 ± 0.565	2.287 ± 0.717
	Obesity	90	10.24 ± 3.419	2.806 ± 0.636	2.339 ± 0.815
		F	4.302	1.963	1.942
		*p*	0.005	0.119	0.122
During the past year, have you suffered a physical activity-related injury? (*N* = 402)	Yes	184	9.49 ± 3.61	2.78 ± 0.62	2.48 ± 0.95
	No	218	10.99 ± 3.61	2.66 ± 0.58	2.42 ± 0.78
		t	−4.14	2.032	0.678
		*p*	0.001	0.043	0.498
How many physical activity-related injuries do you think you have experienced in the last year? (*N* = 184)	Once	93	9.40 ± 3.43	2.65 ± 0.55	2.65 ± 0.95
	Twice	54	10.33 ± 3.84	2.87 ± 0.62	2.34 ± 0.91
	Three or more times	37	8.51 ± 3.51	2.99 ± 0.72	2.26 ± 0.93
		F	2.923	5.032	3.118
		*p*	0.056	0.007	0.047
What type(s) of physical activity-related injury(ies) did you have in the last year? (*N* = 184)	Bruises	43	8.93 ± 3.34	2.71 ± 0.51	2.58 ± 0.84
	Strains	35	11.03 ± 3.42	2.94 ± 0.62	2.29 ± 0.87
	Fractures	22	9.68 ± 2.78	2.49 ± 0.46	2.72 ± 1.38
	Sprains	14	8.79 ± 2.67	2.68 ± 0.58	2.82 ± 0.91
	Concussions	11	7.73 ± 4.45	3.03 ± 0.42	2.86 ± 0.67
	Two or more types	59	9.42 ± 3.99	2.84 ± 0.75	2.27 ± 0.88
		F	2.20	2.145	2.018
		*p*	0.056	0.062	0.078
In which part(s) of your body did physical activity-related injury(ies) occur? (*N* = 184)	Head	17	8.65 ± 3.04	2.81 ± 0.44	2.47 ± 0.99
	Trunk	8	7.75 ± 2.71	2.99 ± 0.41	2.50 ± 0.57
	Upper limbs	26	8.15 ± 3.59	2.84 ± 0.60	2.88 ± 1.05
	Lower limbs	113	10.21 ± 3.53	2.76 ± 0.65	2.48 ± 0.93
	Two or more parts	20	8.60 ± 4.02	2.75 ± 0.73	1.93 ± 0.75
		F	4.126	0.351	3.032
		*p*	0.007	0.843	0.019
Have you had any physical activity-related injuries in the past year that required urgent medical attention? (*N* = 184)	No	141	9.26 ± 3.54	2.73 ± 0.60	2.57 ± 0.93
	Yes	43	10.28 ± 3.76	2.97 ± 0.66	2.18 ± 0.96
		t	−1.637	−2.251	2.376
		*p*	0.103	0.026	0.019

**Table 3 ijerph-20-04610-t003:** Knowledge of physical activity-related injuries, sedentary behaviors, and level of physical activity among Saudi students aged 13–18, stratified by sex × age and sex × BMI categories.

	Groups	*N*	Dependent Variables		
			Correct Answers	Sedentary Behaviors	Physical Activity Score
Boys	13	26	11.38 ± 4.27	2.91 ± 0.54	2.52 ± 0.94
	14	26	10.88 ± 3.41	2.74 ± 0.62	2.40 ± 0.84
	15	28	12.43 ± 4.44	2.76 ± 0.56	2.22 ± 0.71
	16	40	10.00 ± 3.06	2.84 ± 0.70	2.22 ± 0.93
	17	41	9.07 ± 4.10	2.64 ± 0.57	2.49 ± 1.18
	18	45	10.64 ± 3.95	2.87 ± 0.55	2.13 ± 0.62
Girls	13	26	10.23 ± 4.18	2.73 ± 0.57	2.24 ± 0.70
	14	25	10.32 ± 3.49	2.79 ± 0.78	2.62 ± 0.67
	15	28	9.25 ± 3.24	2.58 ± 0.60	2.73 ± 0.93
	16	37	10.08 ± 3.03	2.65 ± 0.59	2.47 ± 0.63
	17	38	10.61 ± 2.92	2.52 ± 0.57	2.65 ± 0.80
	18	42	9.67 ± 3.44	2.62 ± 0.58	2.71 ± 0.92
Total	13	52	10.81 ± 4.22	2.82 ± 0.56	2.38 ± 0.83
	14	51	10.61 ± 3.42	2.76 ± 0.70	2.51 ± 0.76
	15	56	10.84 ± 4.17	2.67 ± 0.58	2.48 ± 0.86
	16	77	10.04 ± 3.03	2.75 ± 0.65	2.34 ± 0.80
	17	79	9.81 ± 3.64	2.58 ± 0.57	2.57 ± 1.01
	18	87	10.17 ± 3.72	2.75 ± 0.57	2.41 ± 0.83
ANOVA	Age (F(*p*))		0.88 (NS)	1.35 (NS)	0.745 (NS)
	Gender and Age (F(*p*))		1.998 (0.027)	1.396 (NS)	2.159 (0.016)
Boys	Underweight	25	11.56 ± 3.318	2.827 ± 0.639	2.280 ± 0.811
	Normal weight	97	9.93 ± 4.231	2.781 ± 0.594	2.381 ± 0.989
	Overweight	24	12.50 ± 4.00	2.723 ± 0.506	2.292 ± 0.820
	Obesity	60	10.43 ± 3.495	2.817 ± 0.624	2.233 ± 0.805
Girls	Underweight	32	9.50 ± 3.059	2.592 ± 0.622	2.445 ± 0.997
	Normal weight	117	9.97 ± 3.436	2.557 ± 0.564	2.671 ± 0.754
	Overweight	17	11.71 ± 3.057	3.017 ± 0.612	2.279 ± 0.565
	Obesity	30	9.87 ± 3.288	2.786 ± 0.670	2.550 ± 0.808
Total	Underweight	57	10.40 ± 3.310	2.695 ± 0.635	2.373 ± 0.916
	Normal weight	214	9.95 ± 3.808	2.659 ± 0.587	2.540 ± 0.878
	Overweight	41	12.17 ± 3.619	2.845 ± 0.565	2.287 ± 0.717
	Obesity	90	10.24 ± 3.419	2.806 ± 0.636	2.339 ± 0.815
ANOVA	BMI (F(*p*))		4.061 (*p* = 0.007)	1.844 (NS)	1.390 (NS)
	Gender and BMI (F(*p*))		2.634 (0.011)	2.592 (0.013)	2.197 (0.034)

**Table 4 ijerph-20-04610-t004:** Logistic regression model for individual predictors of the likelihood of sustaining physical activity-related injuries and those requiring urgent medical attention among Saudi students aged 13–18.

		B (Std. Error)	Wald	Exp(B)	95% CI for Exp(B)	
					Lower	Upper
During the past year, have you suffered a physical activity-related injury?	Gender (1)	1.258 (0.337) ***	13.909	3.520	1.817	6.820
	Correct Answers	−0.136 (0.031) ***	19.506	0.873	0.822	0.927
	Sedentary Behaviors	0.358 (0.183) *	3.829	1.431	0.999	2.048
	Constant	−0.768 (1.269)	0.367	0.464		
Have you had any physical activity-related injuries in the past year that required urgent medical attention?	Gender (1)	1.075 (0.51) *	4.441	2.929	1.078	7.960
	Sedentary Behaviors	0.626 (0.278) *	5.066	1.870	1.084	3.225
	Constant	−4.443 (1.859) *	5.710	0.012		

Note: * *p* < 0.05, *** *p* < 0.001.

**Table 5 ijerph-20-04610-t005:** Multinomial logistic regression model for individual predictors of the number of injuries related to physical activity, their types, and injured body parts, among Saudi students aged 13–18.

			β (Std. Error)	Wald	Exp(β)	95% CI for Exp(β)	
						Lower	Upper
How many physical activity-related injuries do you think you have experienced in the last year?	Once	Intercept	0.037 (1.531)	0.001			
		Correct Answers	−0.133 (0.037) ***	12.702	0.876	0.814	0.942
		(Gender (male))	1.437 (0.407) ***	12.446	4.207	1.894	9.348
	Three or more times	Intercept	−5.954 (2.236)	7.086			
		Correct Answers	−0.231 (0.057) ***	16.320	0.794	0.709	0.888
		Sedentary behaviors	0.822 (0.310) **	7.016	2.274	1.238	4.178
		(Age (14 years))	1.505 (0.687) *	4.797	4.505	1.171	17.320
		(Age (16 years))	1.490 (0.616) *	5.855	4.436	1.327	14.826
What type(s) of physical activity-related injury(ies) did you have in the last year?	Bruises	Intercept	1.211 (2.092)	0.335			
		Fat-free mass	−0.070 (0.036) *	3.852	0.933	0.870	1.000
		Correct Answers	−0.178 (0.050) ***	12.551	0.837	0.759	0.924
		(Gender (male))	1.949 (0.532) ***	13.405	7.024	2.474	19.941
	Strains	Intercept	−6.770 (2.175) **	9.688			
		Sedentary behaviors	0.694 (0.330) *	4.433	2.001	1.049	3.818
		(Gender (male))	1.695 (0.630) **	7.249	5.447	1.586	18.707
	Fractures	Intercept	3.008 (2.87)4	1.096			
		Fat-free mass	−0.125 (0.048) **	6.854	0.883	0.804	0.969
		(Gender (male))	2.200 (0.687) ***	10.257	9.023	2.348	34.672
	Sprains	Intercept	−3.276 (3.652)	0.805			
		Correct Answers	−0.157 (0.081) *	3.795	0.854	0.729	1.001
	Concussions	Intercept	−5.406 (3.962)	1.861			
		Correct Answers	−0.232 (0.087) **	7.053	0.793	0.668	0.941
		Sedentary behaviors	1.370 (0.600) *	5.216	3.936	1.215	12.755
	Two or more types	Intercept	−3.629 (1.815)	3.997			
		Correct Answers	−0.140 (0.044) ***	10.079	0.869	0.797	0.948
		Sedentary behaviors	0.511 (0.258) *	3.917	1.666	1.005	2.763
In which part(s) of your body did physical activity-related injury(ies) occur?	Head or Trunk	Intercept	1.279 (2.923)	0.191			
		Correct Answers	−0.241 (0.066) ***	13.196	0.786	0.690	0.895
		(Age (16 years))	1.780 (0.864) *	4.245	5.930	1.091	32.246
		(Age (17 years))	2.008 (0.858) *	5.474	7.451	1.385	40.080
	Upper limbs	Intercept	−1.000 (2.509)	0.159			
		Correct Answers	−0.215 (0.064) **	11.329	0.806	0.712	0.914
		Sedentary behaviors	0.834 (0.406) *	4.223	2.302	1.039	5.099
		(Gender (male))	2.113 (0.669) **	9.982	8.273	2.230	30.684
	Lower limbs	Intercept	−2.523 (1.462)	2.976			
		Correct Answers	−0.086 (0.035) *	6.055	0.918	0.857	0.983
		(Gender (male))	1.457 (0.392) ***	13.781	4.292	1.989	9.261
		(Age (15 years))	0.789 (0.401) *	3.871	2.202	1.003	4.832
	Two or more parts	Intercept	−0.795 (2.928)	0.074			
		Correct Answers	−0.215 (0.073) **	8.718	0.806	0.699	0.930
		Physical activity score	−0.868 (0.372) *	5.436	0.420	0.202	0.871
		(Age (16 years))	1.686 (0.868) *	3.774	5.398	0.985	29.582

Note: * *p* < 0.05, ** *p* < 0.01, *** *p* < 0.001.

## Data Availability

Data can be provided on request from the corresponding authors.

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
