# Peer review of "Higher Sedentary Behaviors and Lower Levels of Specific Knowledge Are Risk Factors for Physical Activity-Related Injuries in Saudi Adolescents"

_ijerph, 2023, doi:10.3390/ijerph20054610_

Round 1
Reviewer 1 Report
Title: Higher Sedentary Behaviors and Lower Levels of Specific Knowledge Are Risk Factors for Physical Activity-Related Injuries in Saudi Adolescents
Article Type: Article
Summary
In this study, the authors aimed to examine the frequency, location, type, and severity of PA-related injuries in Saudi students aged 13 to 18 years as well as to identify associated risk factors. Participants were 402 students (206 boys and 196 girls) who participated in this study. The results indicated that better specific knowledge was associated with a lower likelihood of sustaining injuries but increased sedentary behaviors were associated with a greater likelihood of sustaining a PA-related injury.
Evaluation
The topic of this study was interesting for me and even for publication in the Journal. The design for the study was appropriate to answer the research questions. For an original scientific paper, the manuscript was quite straightforward, therefore, it seems that this manuscript does meet the conditions for publication in the Journal.
Minor
Please add age mean to the abstract for boys and girls.
Please add significant level for each result in the abstract.
How you calculated the sample size? Add this information to the method section.
L 167, Please write PA instead of AP.
L 204, please write the year in the ().
Please add the effect size for each comparison in table 1.
In the results section, please write the statistics for One-way ANOVA for all comparison, you wrote only significant level but not F index, df, ….
Author Response
Responses point-by-point
The authors are very grateful to Reviewer 1 for his valuable comments and suggestions, which have been incorporated into the revised edition. All comments were addressed point by point and the changes are highlighted in yellow. New references were added to the reference list and the detailed responses and corresponding changes are as follows:
Reviewer 1
In this study, the authors aimed to examine the frequency, location, type, and severity of PA-related injuries in Saudi students aged 13 to 18 and identify associated risk factors. The participants were 402 students (206 boys and 196 girls) who participated in this study. The results indicated that better specific knowledge was associated with a lower likelihood of sustaining injuries but increased sedentary behaviors were associated with a greater likelihood of sustaining a PA-related injury.
Response: We appreciate your feedback; your comments are very important. Thank you for your time in evaluating our work.
Evaluation
The topic of this study was interesting for me and even for publication in the Journal. The design for the study was appropriate to answer the research questions. For an original scientific paper, the manuscript was quite straightforward, therefore, it seems that this manuscript does meet the conditions for publication in the Journal.
Minor
Please add age mean to the abstract for boys and girls.
Response: added.
Please add significant level for each result in the abstract.
Response: Significant levels have been added where possible.
How you calculated the sample size? Add this information to the method section.
Response: This information has been added right at the beginning of the Participants section
L 167, Please write PA instead of AP.
Response: Corrected.
L 204, please write the year in the ().
Response: This reference is missing. It has been added to the list of references and its number has been put in brackets.
Please add the effect size for each comparison in table 1.
Response: The effect size has been added and the necessary changes have also been made.
In the results section, please write the statistics for One-way ANOVA for all comparisons, you wrote only significant level but not F index, df, ….
Response: The F index and the df have been added.

Reviewer 2 Report
Dear authors,
I enjoyed reading your paper. However the introduction needs to be improved by;
1. Ensuring that the content aligns with the age group, adolescents rather than college students.
2. Using recent references i.e. Line 63 reference 12, the publication date is 2011.
3. Be more specific about the types of injuries i.e. acute or chronic and provide examples.
In the Methods section, provide the reference for "NSW Schools Physical Activity and Nutrition Survey".
In the results section, Figure 1, (B), the labels for the x-axis are in bold.
In the discussion, the use of the word "proficiency" on line 297 does not make sense. The whole sentence from lines 295 to 298 needs to be reworded.
Author Response
Responses point-by-point
The authors are very grateful to Reviewer 2 for his valuable comments and suggestions, which have been incorporated into the revised edition. All comments were addressed point by point and the changes are highlighted in yellow. New references were added to the reference list and the detailed responses and corresponding changes are as follows:
Reviewer 2
Comments and Suggestions for Authors
Dear authors,
I enjoyed reading your paper. However, the introduction needs to be improved by;
- Ensuring that the content aligns with the age group, adolescents rather than college students.
Response: Necessary changes have been made
- Using recent references i.e. Line 63 reference 12, the publication date is 2011.
Response: Wherever possible, older references have been removed or replaced with newer ones throughout the manuscript.
- Be more specific about the types of injuries i.e. acute or chronic and provide examples.
Response: PA-related injuries include both acute and chronic injuries. A paragraph, with examples, has been added on this topic.
In the Methods section, provide the reference for "NSW Schools Physical Activity and Nutrition Survey".
Response: Provided.
In the results section, Figure 1, (B), the labels for the x-axis are in bold.
Response: Corrected.
In the discussion, the use of the word "proficiency" on line 297 does not make sense. The whole sentence from lines 295 to 298 needs to be reworded.
Response: Changed as follows “The one-way ANOVA test showed that levels of knowledge about PA-related injuries were almost similar across all age groups. However, significant differences were noted among participants stratified by BMI categories (f = 4.302; df = 398; p=0.005), demonstrating higher competence among overweight participants compared to their normal-weight (p=0.002) and obese (p=0.027) peers”.

Reviewer 3 Report
The authors have chosen a relevant and important topic for their research especially that physical activity is a right for everyone and it is surprising that the physical education program in girls' schools was approved only in 2017 in Saudi society.
I enjoyed reading it and welcome the straightforward methodology operationalization. My major concern is about the procedure and the assumptions regarding differences between girls and boys. They can be obvious since girls have PA in the school curriculum only for 5 years.
Please provide more details on the procedures (when the study was conducted). The description is far too simple and makes it difficult to understand the dynamics. The transition to the importance of interventions is not clear to me.
Please corect at line 204 "Beaton et al. ()".
It is necessary to improve the conclusions and relate them to the aim of the research.
Author Response
Responses point-by-point
The authors are very grateful to Reviewer 3 for his valuable comments and suggestions, which have been incorporated into the revised edition. All comments were addressed point by point and the changes are highlighted in yellow. New references were added to the reference list and the detailed responses and corresponding changes are as follows:
Reviewer 3
The authors have chosen a relevant and important topic for their research especially that physical activity is a right for everyone and it is surprising that the physical education program in girls' schools was approved only in 2017 in Saudi society.
I enjoyed reading it and welcome the straightforward methodology operationalization. My major concern is about the procedure and the assumptions regarding differences between girls and boys. They can be obvious since girls have PA in the school curriculum only for 5 years.
Response: We appreciate your feedback; your comments are very important. We thank you for the time you have taken to evaluate our work. Indeed, the assumptions may be obvious because of the current situation of girls burdened by religion and tradition. We hope that this will change in the future.
Please provide more details on the procedures (when the study was conducted). The description is far too simple and makes it difficult to understand the dynamics. The transition to the importance of interventions is not clear to me.
Response: The necessary information has been added to the "Procedures and Ethics" section and some modifications have been made to make it clearer.
Please correct at line 204 "Beaton et al. ()".
Response: Corrected.
It is necessary to improve the conclusions and relate them to the aim of the research.
Response: The necessary changes have been made to make the conclusions clearer.

Round 2
Reviewer 3 Report
I believe that the manuscript in its revised form meets the requirements to be published in IJERPH and I agree with the current form, which is improved from a qualitative point of view.